# Influence of Shield Attitude Change on Shield–Soil Interaction

**Xiang Shen** [1,2,3] **, Da-Jun Yuan** [1,2,3] **and Da-Long Jin** [1,2,3,]*

1   Key Laboratory of Urban Underground Engineering of Ministry of Education, Beijing Jiaotong University,
    Beijing 100044, China; xshenbjtu@126.com (X.S.); yuandj603@163.com (D.-J.Y.)
2   Tunnel and Underground Engineering Research Center of Ministry of Education, Beijing Jiaotong University,
    Beijing 100044, China
3   School of Civil Engineering, Beijing Jiaotong University, Beijing 100044, China
*   Correspondence: 14115317@bjtu.edu.cn or jindalong@163.com; Tel.: +86-1800-109-1006



**Featured Application: Shield attitude control and shield tunnel construction risk control.**

**Abstract:** The mechanism of shield–soil interaction and multi-phase equilibrium control theory in shield tunneling process still lack sufficient understanding. Aiming at this problem, with the improved calculation method of loose earth pressure, the initial boundary problem of shield attitude calculation was solved. Based on the ground reaction curve, the shield–soil interaction was simulated by the equivalent springs, and the displacement of surrounding soil was calculated during the change of the shield attitude. Then, the theoretical method of surrounding soil load acting on the shield were obtained. In summary, the calculation method of shield attitude was obtained. This method has three main applications in engineering, namely the inversion of shield–soil interaction force, the prediction of pitch angle and the prediction of yawing angle. Finally, combined with Jinan Metro Line R2 shield tunnel project, the shield attitude was monitored in real time and compared with the theoretical value. The results show that the trend of the theoretical values of pitch angle and yawing angle were basically the same as the measured value, but the theoretical value was generally larger than the measured value. The research results can provide a useful reference for the shield attitude adjustment.

**Keywords:** shield; shield attitude; pitch angle; yawing angle; interaction; advance cylinders

## 1. Introduction

Shield tunneling has been widely used in the construction of urban subway tunnels. Due to the uncertainty of load and stratum during the excavation process, the shield cannot always be consistent with the theoretical design axis [1–4]. The automatic control system used to control the shield attitude is based on existing engineering experience and not supported by appropriate theories [5]. Therefore, how to properly control the advance load is one of the main concerns in the construction of shield tunnels. At present, relevant research mainly focuses on the two directions of advance load modeling and automatic correction.

The difficulty in establishing a shield tunneling mechanics model is that the forces around the shield cannot be calculated and expressed accurately. Therefore, many scholars have carried out a series of studies on the establishment of shield tunneling mechanics models and attitude control. Koyama et al. [6] mentioned that it is impossible to deal with the precise control of the shield machine in the case of complex geological structures and sharp curves. Xie et al. [7] applied the automatic trajectory tracking control system to the control of the shield's route, but these systems also only rely on empirical relationships, without precise theoretical background. Shimizu et al. [8] analyzed the tunneling and motion laws of

tunnel construction without the influence of nonlinear factors. Shimizu et al. [9] established a mathematical analysis model of shield tunneling movement. Robust theory was applied to shield attitude control by Komiya et al. [10]. Based on the force balance condition of the shield, Sugimoto et al. [11] determined the relationship between the advance load and the earth pressure, and then connected the earth pressure with the shield attitude, thus establishing the theoretical model of the shield advance load. Yue et al. [12] and Sun et al. [13] proposed a shield attitude and trajectory automatic control system based on the sliding mode robust controller and a shield attitude dynamic coordination control system based on the dynamic model of the shield attitude adjustment process. Ates et al. [14] believed that it is crucially important to select a proper TBM and define its basic specifications such as installed cutter-head torque and TBM thrust capacities. Zhang et al. [15] established a predicting model of the thrust and torque for the total load that fully reflects the influences of geological, operating, and structural parameters. In summary, the current research on shield attitude is more about the application of control theory, while ignoring the basic problems such as shield–soil interaction and shield tunneling mechanics.

With the necessity to accurately predict performance of machines in different ground conditions, many researchers have worked to develop new prediction models or adjustment factors for the common existing models [16]. Acaroglu et al. [17] established a model to predict specific energy requirement of constant cross-section disc cutters in the rock cutting process by using fuzzy logic method. Barton et al. [18] proposed a model named $Q_{TBM}$ that uses many input parameters (such as RQD, joint condition, Stress condition, intact rock strength, quartz content and TBM thrust). The modified CSM model was added rock mass properties as input parameters into the model [19]. In hard rock formations, such models have often shown results that it would not expand too much. However, in the soft soil layer, the interaction model for shield tunneling is still in its infancy.

In view of the above deficiencies, the influence of the shield attitude parameter on the shield–soil interaction was studied. The shield–soil interaction was represented by an equivalent spring, and the interaction force between the shield and the soil was solved and analyzed according to the ground reaction curve. Based on this, the theoretical calculation method of shield attitude was proposed and compared with the actual project.

## 2. Methodology

### 2.1. Model of Load Acting on Shield

Based on the shield load model established by Sugimoto [20], the contact relationship between the shield and the stratum is blurred. The force of shield–soil interaction is decomposed along the axis direction, the normal direction, and the tangential direction of the shield, and the load model is shown in Figure 1.

The loads acting on the shield are: $f_1$, shield self-weight; $f_2$, force on the shield tail; $f_3$, advance load provided by the jacks; $f_4$, load acting on the cutter head; and $f_5$, surrounding soil load acting on the shield periphery [21].

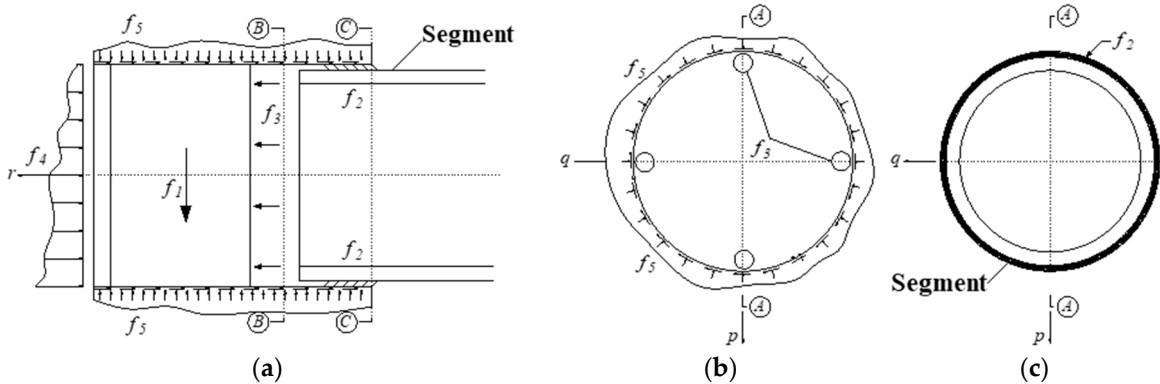

**Figure 1.** Shield load model: (**a**) Section A-A; (**b**) Section B-B; and (**c**) Section C-C.

If the shield is regarded as a rigid body, the distance between any two points on the shield remains unchanged during the tunneling process. Six parameters are required to describe the shield attitude: cutterhead center coordinate $(x_0, y_0, z_0)$, yawing angle $\alpha$, pitch angle $\beta$, and rolling angle $\Omega$. The attitude angles are shown in Figure 2a.

The coordinate system is established as shown in Figure 2b: in the geodetic coordinate system $C^E$, the x-axis is vertically downward and the y-axis is on the horizontal plane; in the shield-structure coordinate system $C^M$, the p-axis is vertically downward and the r-axis is along the shield axis; and the coordinate system $C^M$ is rotated by a certain angle around the r-axis to obtain a coordinate system $C^{MR}$.

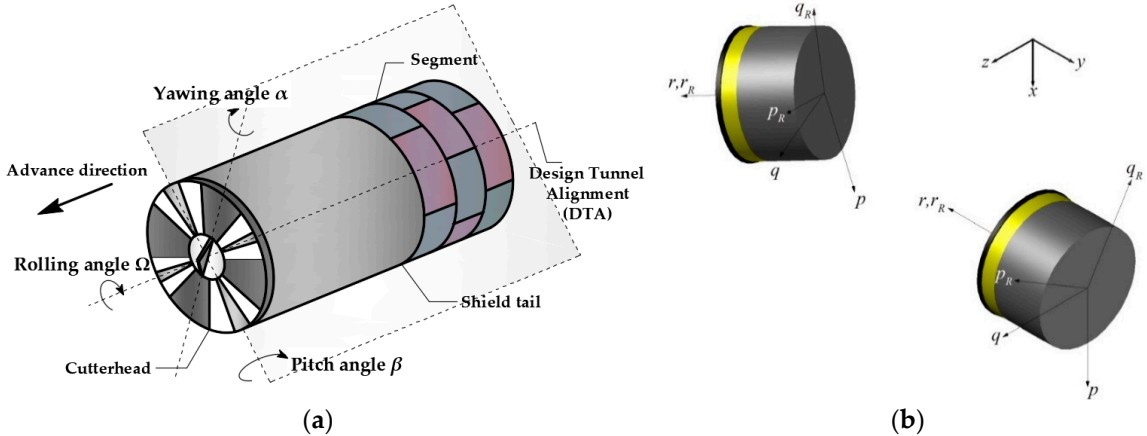

(**a**)                                                                                                (**b**)

**Figure 2.** Attitude angles and coordinate system: (**a**) attitude angles; and (**b**) coordinate system.

When the shield is considered to be in a static equilibrium state during the tunneling process, the static balance equation must be met:

$$
\begin{aligned}
\sum_{n=1}^{5} f_n^b &= 0 \\
\sum_{n=1}^{5} M_n^b &= 0
\end{aligned}
\tag{1}
$$

Then:

$$
\begin{aligned}
F_{1e}^b + F_{2e}^b + F_{3e}^b + F_{4e}^b + F_{5e}^b &= 0 \\
M_{1e}^b + M_{2e}^b + M_{3e}^b + M_{4e}^b + M_{5e}^b &= 0
\end{aligned}
\tag{2}
$$

where $e = p$, $q$, and $r$ represent different directions of force; and $b = E$, $M$, and $MR$ are expressed in different coordinate systems.

### 2.2. Initial Earth Pressure Calculation

One of the keys to establishing a shield–soil interaction model during the change of shield attitude is the calculation of initial earth pressure. Loose earth pressure theory is often used for deformation analysis of segments. The initial shield stress state should also be in the non-uniform state. This part solves the initial boundary problem of the shield interaction model by means of relevant theories.

#### 2.2.1. Calculation of Loose Soil Pressure in the Overlaying Soil

Terzaghi deduced the classical theory of loose earth pressure through the trap-door test [22]. The analytical expression of the vertical stress $\sigma_v$ of the trap-door is:

$$
\sigma_v = \frac{B_1 \gamma - c}{K_0 \tan \varphi} \left( 1 - e^{-\frac{K_0 \tan \varphi H}{B_1}} \right) + P_0 e^{-\frac{K_0 \tan \varphi H}{B_1}}
\tag{3}
$$

where $\sigma_v$ is the loose earth pressure; $K_0$ is the later pressure coefficient; $H$ is the overlying soil thickness; $P_0$ is the upper load; $c$ is the cohesion; $\varphi$ is the internal friction angle; $\gamma$ is the weight density; and $B_1$ is the loose band width.

$$B_1 = R\cot\left(\frac{\pi/4 + \varphi/2}{2}\right) \tag{4}$$

where $R$ is the shield radius.

Li [23] introduced the factor of formation loss rate based on the stress of Terzaghi loose soil, and set the key parameter $A_1$ to correct the theoretical solution of Terzaghi loose soil pressure:

$$\sigma_v = \frac{B_1\gamma - c}{A_1}\left(1 - e^{-\frac{A_1 H}{B_1}}\right) + P_0 e^{-\frac{A_1 H}{B_1}} \tag{5}$$

$$A_1 = \frac{1 + K_a\tan^2\theta}{\tan^2\theta + K_a} \cdot \frac{(1 - K_a)\tan\theta}{1 + K_a\tan^2\theta} \tag{6}$$

where $K_a$ is the active side pressure coefficient, $K_a = \tan^2(45° - \varphi/2)$.

According to Figure 3, the following conclusions can be drawn:

$$\tan\theta = \frac{B_1}{2s} - \frac{s}{2B_1} \tag{7}$$

where $s$ is the displacement of the dome.

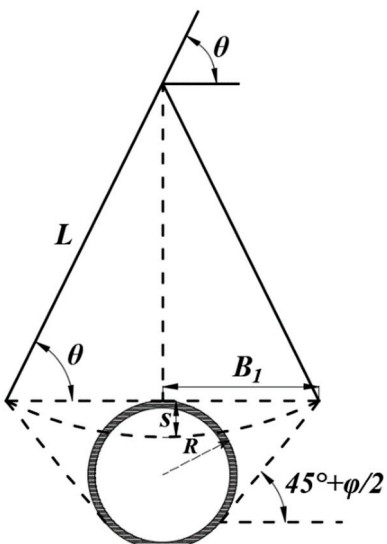

**Figure 3.** Calculation of tanθ [23].

### 2.2.2. Initial Force Acting on the Shield Periphery

The earth pressure around the shield increases with the increase of the depth of the soil, as shown in Figure 4. $\eta$ is the angle with the $p$-axis in the $C^M$ coordinate system. To simplify the calculation, linear regression method is adopted. In the process of $\eta$ from $\pi/2$ to $3\pi/2$, the corresponding normal earth pressure acting on the shield at any point is linearly increased, so that the normal force on the shield can be obtained. The normal earth pressure at any point in the shell is shown in Figure 5.

Therefore, the earth pressure on the shield shell can be calculated as follows:

$$p_\eta = \begin{cases} \frac{p_1 - p_2}{\pi}\left(\eta + \frac{\pi}{2}\right) + p_2 & 0 \leq \eta \leq \frac{\pi}{2} \\ \frac{p_2 - p_1}{\pi}\left(\eta - \frac{\pi}{2}\right) + p_1 & \frac{\pi}{2} < \eta \leq \frac{3\pi}{2} \\ \frac{p_1 - p_2}{\pi}\left(\eta - \frac{3\pi}{2}\right) + p_2 & \frac{3\pi}{2} < \eta \leq 2\pi \end{cases} \tag{8}$$

$$p_2 = p_1 + \frac{4F_{1p}^M}{\pi DL}$$ (9)

where $p_1$ is the normal earth pressure at the top of the shield, calculated according to improved loose soil pressure formula [23]; $p_2$ is the normal earth pressure at the bottom of the shield; $D$ is the shield diameter; and $L$ is the shield length.

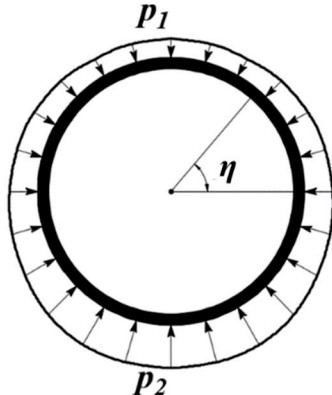

**Figure 4.** Initial shield shell force calculation model.

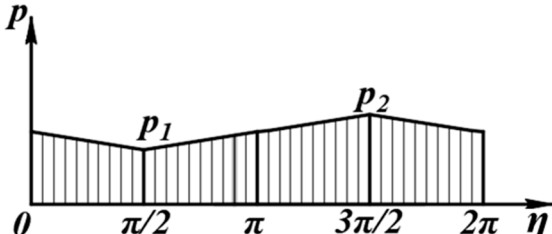

**Figure 5.** Initial shield shell force distribution curve.

*2.3. Shield Shell–Soil Interaction Model*

2.3.1. Basic Assumption

The shield is constrained by the soil. Based on the characteristics of the ground reaction curve, the soil can be approximated as an equivalent spring within a certain range (Figure 6). The equivalent spring is nonlinear deformation, and the correlation coefficient can be obtained from the ground reaction curve [24]. According to the deformation of the soil spring, the interaction force between the shield and the soil can be obtained.

The center of gravity of the shield machine is generally not in the geometric centroids, as shown in Figure 7a. Point $S$ is the center of gravity of the shield, Point $O$ is the geometric centroid, and $l_s$ is the distance between two points.

When the shield attitude is adjusted in the actual project, the yawing angle and pitch angle should be changed simultaneously. However, when the time interval of shield attitude adjustment tends to infinitesimal, it can be considered that the pitch angle and yawing angle are in succession. The order in which the pitch and yawing angles occur does not affect the final calculation result from the aspect of the mechanics analysis. Assuming that the shield is performing attitude adjustment, it can be divided into four stages, as follows:

(i)   Stage 1: Gravity stage. The point of gravity action is moved to the geometric centroid while the gravity eccentric moment $M_G$ is generated, and only gravity is considered at this stage. Under the action of gravity, the shield machine produces a displacement of $\Delta s_v$ in the vertical direction, and $\Delta s_v$ is positive downward and negative upward, as shown in Figure 7b.

(ii)   Stage 2: Gravity eccentric moment stage. The shield deflects due to gravity eccentric bending moment on the vertical plane. The pitch angle caused by the $M_G$ is called $\beta_G$, as shown in Figure 7c. The angle is positive when its direction is counterclockwise, and the angle is negative when its direction is clockwise.

(iii)  Stage 3: Vertical bending moment stage. Upper and lower partition jacks produce deflection moments $M_A^\beta$ on the vertical plane. The pitch angle caused by the $M_A^\beta$ is called $\beta_A$, as shown in Figure 7c. The angle is positive when its direction is counterclockwise, and the angle is negative when its direction is clockwise.

(iv)  Stage 4: Horizontal bending moment stage. Under the action of horizontal deflection bending moment $M_A^\alpha$, $\alpha$ which is called yawing angle is generated on the horizontal plane, as shown in Figure 7d. $\alpha$ is positive when its direction is counterclockwise and negative when its direction is clockwise.

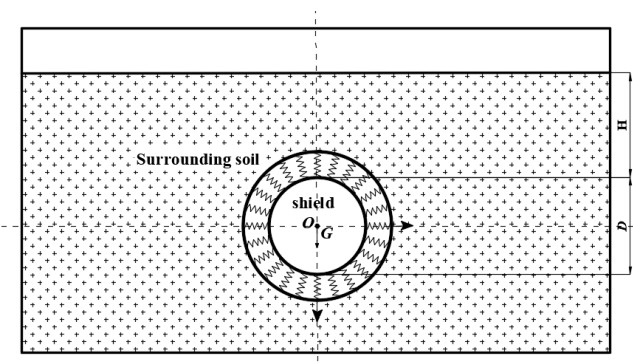

**Figure 6.** Shield–soil interaction model.

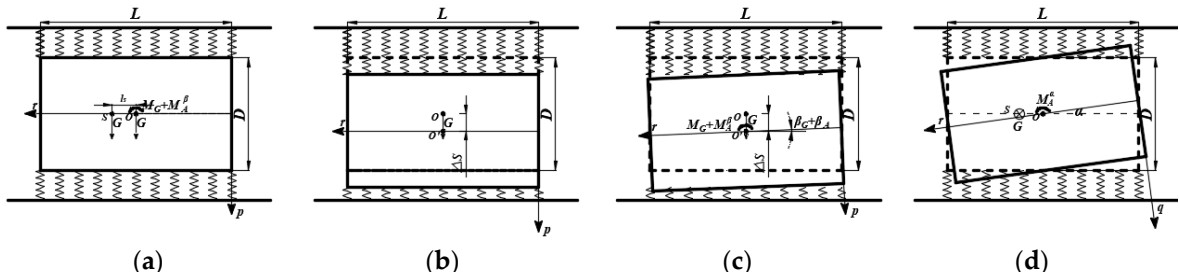

(**a**)               (**b**)               (**c**)               (**d**)

**Figure 7.** Shield attitude change process: (**a**) Stage1; (**b**) Stage2; (**c**) Stage3; and (**d**) Stage 4.

### 2.3.2. Geometric Parameters

Assuming that the section of the shield at $r = L$ has a displacement $\Delta s_v$ in the vertical direction and $\Delta s_h$ in the horizontal direction, the displacement occurring at each point on the shield is calculated. As shown in Figure 8, $D$ is the shield diameter, $r$ is the shield radius, $L$ is the length of the shield, $O$ is the initial position of the shield, $O'$ is the final position of the shield, and $\eta$ is the angle with the $p$-axis in the $C^M$ coordinate system.

As shown in Figure 8, the following inference can be made:

$$
\begin{aligned}
&\text{Circle } O: \ x^2 + y^2 \ = \ r^2 \\
&\text{Circle } O: \ (x - \Delta s_h)^2 + (y - \Delta s_v)^2 = r^2
\end{aligned}
\tag{10}
$$

Polar coordinate conversion: $x = \rho \cos(-\eta)$; $y = \rho \sin(-\eta)$. Bring in Equation (4) to get:

$$
\begin{aligned}
&\text{Circle } O: \ \rho = r \\
&\text{Circle } O: \ \rho^2 - 2(\cos\eta \Delta s_h - \sin\eta \Delta s_v)\rho + \Delta s_h^2 + \Delta s_v^2 = r^2
\end{aligned}
\tag{11}
$$

$$\overline{MM'} = \cos\eta \cdot \Delta s_h - \sin\eta \cdot \Delta s_v + \sqrt{r^2 - \sin 2\eta \cdot \Delta s_h \cdot \Delta s_v - \sin^2\eta \cdot \Delta s_h^2 - \cos^2\eta \cdot \Delta s_v^2} - r \tag{12}$$

$\overline{MM'} < 0$ indicates that the shield is subjected to active earth pressure, and $\overline{MM'} > 0$ indicates that the shield is subjected to passive earth pressure. Therefore, the vertical and horizontal displacement of the shield:

$$U_v(\theta) = \overline{MM'} \cdot |\sin\eta|, U_h(\theta) = \overline{MM'} \cdot |\cos\eta| \tag{13}$$

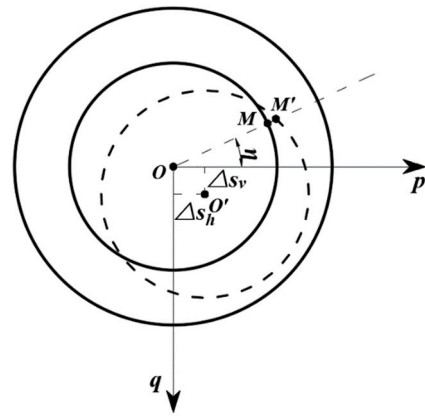

**Figure 8.** Displacement calculation diagram.

### 2.3.3. Solution of Vertical and Horizontal Forces of Shield–Soil Interaction

Through the ground reaction curve (Figure 9), considering the change of the shield attitude, the vertical and horizontal interaction forces between the shield and the soil are deduced and analyzed under the premise of small deformation.

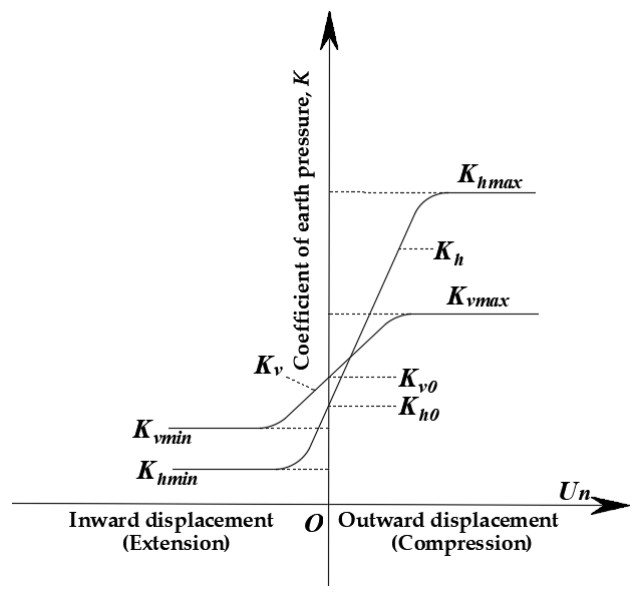

**Figure 9.** Ground reaction curve [24].

The ground reaction curve [24], which shows the relationship between the ground displacement and the earth pressure acting on the shield periphery as shown in Figure 9, can be represented by

$$K_i(U_i) = \begin{cases} (K_{io} - K_{imin})\tanh\left[\frac{a_i U_i}{K_{io} - K_{imin}}\right] + K_{io} & (U_i \le 0) \\ (K_{io} - K_{imax})\tanh\left[\frac{a_i U_i}{K_{io} - K_{imax}}\right] + K_{io} & (U_i \ge 0) \end{cases} \tag{14}$$

where $K$ is the coefficient of earth pressure, which is defined as the earth pressure $\sigma$ divided by the initial vertical earth pressure $\sigma_{v0}$ ($K_i = \sigma_i/\sigma_{v0}$); $U$ is the ground displacement; $a$ is the gradient of functions $K(U)$, which represents the coefficient of subgrade reaction $a_i$; subscripts $v$ and $h$ are the vertical horizontal directions, respectively; and subscripts $o$, *min*, and *max* are the initial, lower limit and upper limit of the coefficient of earth pressure, respectively.

The shield is discretized, divided into $n$ parts along the circumferential direction, and divided into $m$ parts along the longitudinal direction, as shown in Figure 10.

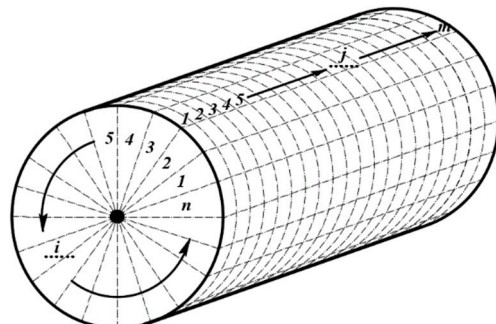

**Figure 10.** Shield discretization.

Vertical and horizontal stress of each unit are obtained as follows:

$$\sigma_{vsij} = K_v\big[U_{vij}(\eta)\big]\sigma_{v0ij}, \sigma_{hsij} = K_h\big[U_{hij}(\eta)\big]\sigma_{h0ij} \tag{15}$$

where $\sigma_{v0ij}$ and $\sigma_{h0ij}$ are the initial vertical and horizontal earth pressures, respectively, and the initial earth pressure is calculated as the loose earth pressure.

Therefore, the conclusions can be obtained as follows:

$$F_{5p}^{M} = \sum_{i=1}^{n}\sum_{j=1}^{m}\sigma_{vsij}A_{ij}, F_{5q}^{M} = \sum_{i=1}^{n}\sum_{j=1}^{m}\sigma_{hsij}A_{ij} \tag{16}$$

where $A_{ij}$ is the area of each unit of the shield shell, and $A_{ij} = \frac{2\pi}{n}{\cdot}r{\cdot}\frac{L}{m}$.

It can be known from Equations (1) and (2):

$$F_{5p}^{M} + F_{1p}^{M} = 0, F_{5q}^{M} = 0 \tag{17}$$

2.3.4. Solution of Bending Moment of Shield–Soil Interaction

Assuming that the pitch angle of the shield is $\beta$ ($\beta = \beta_G + \beta_A$) and the yawing angle is $\alpha$, then the vertical and horizontal displacement of each element on the shield are determined as follows:

$$\begin{aligned}\Delta s_v' &= \Delta s_v + \left(x_r^M - \tfrac{L}{2}\right)\tan\beta \\ \Delta s_h' &= \left(x_r^M - \tfrac{L}{2}\right)\tan\alpha\end{aligned} \tag{18}$$

where $\Delta s_v'$ and $\Delta s_h'$ are the vertical and horizontal displacements of the elements on different sections, respectively. $x_r^M$ is the value of the r axis in the $C^M$ coordinate system.

$$M_{5q}^{M} = \sum_{i=1}^{n}\sum_{j=1}^{m}\sigma_{vsij}\left(x_{rij}^{M} - \frac{L}{2}\right)A_{ij}, M_{5p}^{M} = \sum_{i=1}^{n}\sum_{j=1}^{m}\sigma_{hsij}\left(x_{rij}^{M} - \frac{L}{2}\right)A_{ij} \tag{19}$$

It can be known from Equations (1) and (2):

$$M_{5q}^{M} + M_{1q}^{M} + M_A^{\beta} = 0, M_{5p}^{M} + M_A^{\alpha} = 0 \tag{20}$$

## 3. Shield Pitch Angle and Yawing Angle Calculation Method

### 3.1. Solution Process of Yawing Angle and Pitch Angle

According to the above calculation theory, the calculation method of the pitch angle and yawing angle of the shield can be summarized. The specific calculation process is shown in Figure 11.

The first stage is to input the basic parameters such as soil parameters and shield parameters, and establish the shield–soil interaction model. The relationship curve between the vertical displacement and vertical force on the shield (RCDV) is obtained. Through the RCDV, the vertical displacement $\Delta s_v$ under gravity is obtained. In the second stage, the $\Delta s_v$ obtained in the first stage and the unknown pitch angle are set into Equation (18), and the relationship curve between the pitch angle and moment on the vertical plane (RCPV) is obtained, which is obtained under the action of gravity eccentric bending moment. Through the RCPV, the pitch angle $\beta_G$ generated by the $M_G$ and the $\beta_A$ generated by the $M_A^\beta$ can be obtained. In the third stage, $\Delta s_v$, $\beta_G$, and $\beta_A$ are brought into the calculation program, and the unknown yawing angle value is continuously given. The relationship curve between the yawing angle and horizontal bending moment applied by the shield (RCYH) is obtained. Through the RCYH, the $M_A^\alpha$ is calculated, then the yawing angle $\alpha$ is obtained. Finally, $\Delta s_v$, $\beta_G$, $\beta_A$, and $\alpha$ are output.

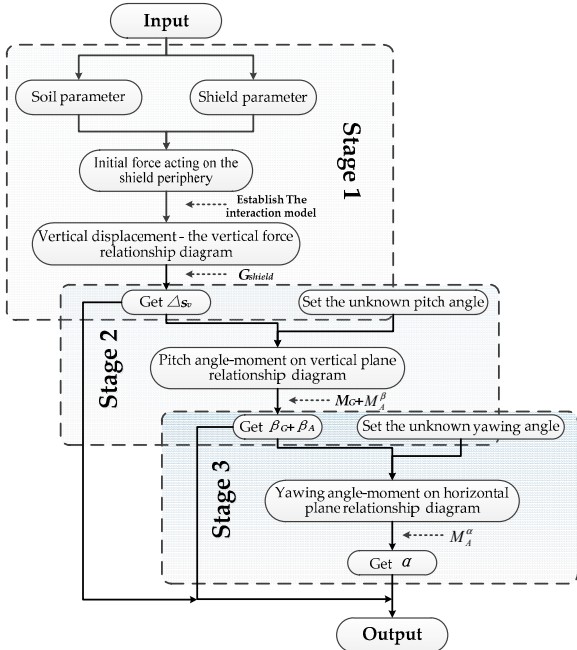

**Figure 11.** Shield attitude calculation flow chart.

### 3.2. Calculation Example

Based on the above derivation results, the following set of typical parameters for calculation was selected. Suppose a shield tunnel is excavated in a uniform sand layer. The soil weight $\gamma$ is 19 kN/m³, the cohesion $c$ is 0 kPa, and the internal friction angle $\varphi$ is 30°. The shield diameter $D$ is 12 m, the length $L$ is 10.8 m, the shield mass $G_{shield}$ is 3000 t, and the thickness of covering soil is 24 m. The displacement of the dome is 5 mm, and the eccentricity of gravity $l_s$ is 3.5 m. $M_A^\beta$ = 20 MN·m and $M_A^\alpha$ = 75 MN·m. The ground reaction curve coefficient is shown in Table 1 [11].

Through the Matlab programming calculation, the shield attitude parameters could be calculated. As shown in Figure 12a, the weight of the shield machine was 3000 t, and $\Delta s_v$ was obtained ($\Delta s_v$ = 8.1 mm). In the second stage, the sum of $M_G$ and $M_A^\beta$ is 125 MN·m, then the pitch angle $\beta$ could be obtained (Figure 12b) ($\beta$ = 1.16°). In the third stage, it was found that $M_A^\alpha$ = 75 MN·m, and then the yawing angle $\alpha$ could be obtained (Figure 12c) ($\alpha$ = 0.7°).

**Table 1.** Ground reaction curve coefficient table [11].

| $K_{hmin}$ | $K_{ho}$ | $K_{hmax}$ | $K_h$ | $K_{vmin}$ | $K_{vo}$ | $K_{vmax}$ | $K_v$ |
|------------|----------|------------|-------|------------|----------|------------|-------|
| 0.3 | 1 | 5 | 3 MN/m$^3$ | 0.3 | 1 | 5 | 3 MN/m$^3$ |

(a)

(b)

(c)

**Figure 12.** Calculation of shield attitude parameters: (**a**) Stage 1, solving $\Delta s_v$; (**b**) Stage 2, solving $\beta$; and (**c**) Stage 3, solving $\alpha$.

## 4. Application Engineering

### 4.1. Engineering Details

Jinan is located in the central and western parts of Shandong Province, overlooking the Mount Tai to the south and across the Yellow River to the north, as shown in Figure 13. Jinan is famous for its springs with complex groundwater network [25]. The underground spring water is extremely precious for the city of Jinan, thus the protection of spring water during the construction of the subway is very important. Shield attitude control is the key to spring protection during shield construction. Therefore, Jinan Metro Line R2 Line was analyzed and discussed.

Jinan Metro Line R2 is an east–west city express line. It is a rail transit backbone line that alleviates east–west traffic pressure and supports the expansion of strip-shaped urban space. This study selected the Wangfuzhuang–Renjiazhuang section of Jinan Metro Line R2. The buried depth of the tunnel varies from 7.9 to 15.2 m. The maximum gradient of the tunnel is 12‰, and it passes underneath important regions such as Beijing–Shanghai high-speed railway, Beijing–Fuzhou expressway and Yufu River. The advancement of shield machine usually encountered a silty clay soil and loess during the tunneling process in the studied section. The geological profile of the encountered stratum is displayed in Figure 14. The construction process of Ring 504 segments was selected for analysis and research. The soil characteristic parameters of Ring 504 are shown in Figure 15. The third and fourth layers are the soil parameters of the shield excavation section.

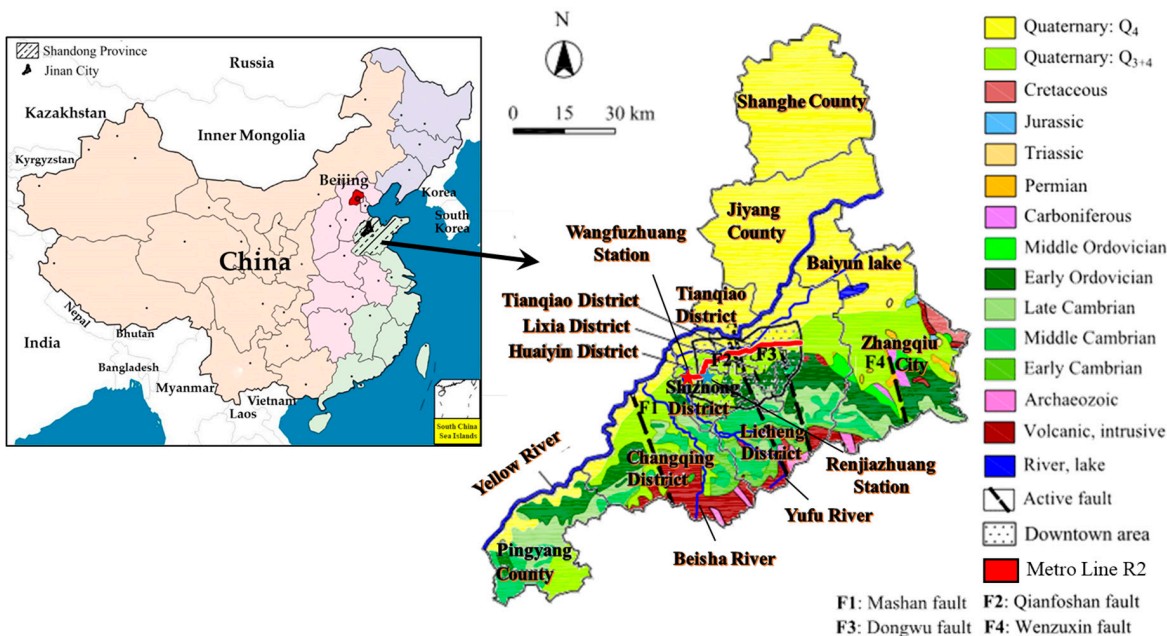

**Figure 13.** Plan view of the district division and geology in Jinan [26].

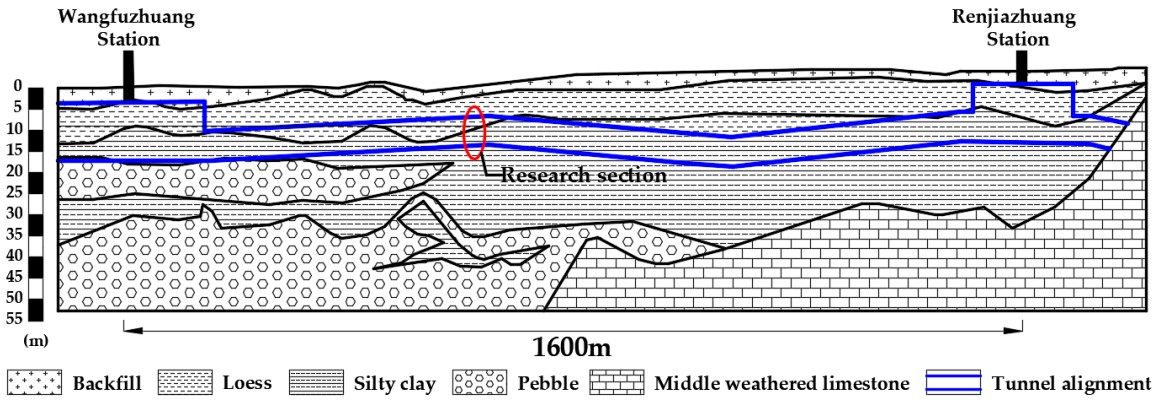

**Figure 14.** Geology section of Wangfuzhuang–Renjiazhuang section of Jinan Metro Line R2.

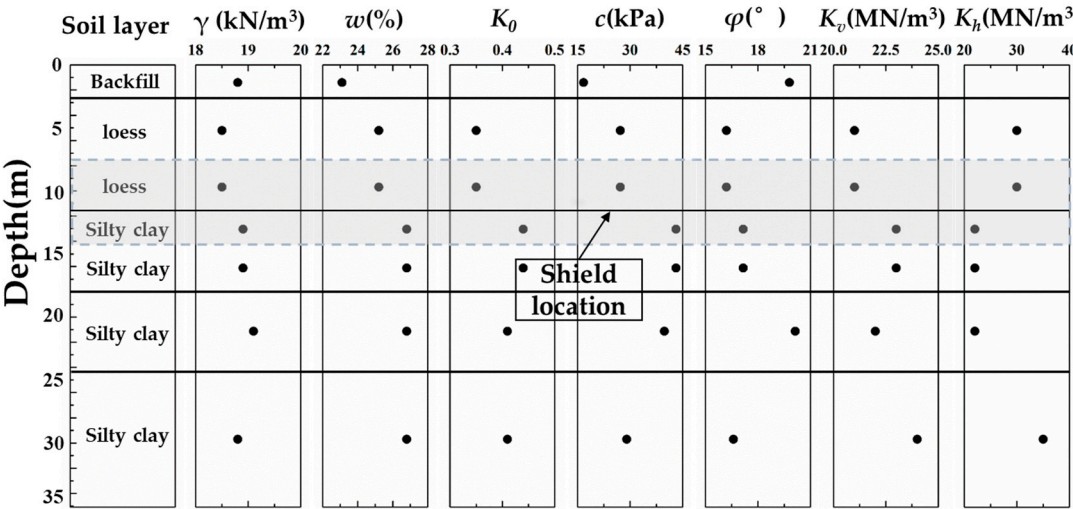

**Figure 15.** Geotechnical profiles of Ring 504. γ, weight density; ω, water content; $K_0$, later pressure coefficient; $c$, cohesion; $φ$, internal friction angle; $K_v$, vertical foundation coefficient; $K_h$, horizontal foundation coefficient.

### 4.2. Shield Details

An earth pressure balanced shield (EPB) machine of 6.68 m in diameter was used to excavate in Jinan Metro Line R2. The length of shield is 9 m, and its weight is about 500 t. Cutter opening ratio is 35%, as shown in Figure 16a. The shield machine specifications are summarized in Table 2. The shield machine advance system uses 22 sets of cylinders, which are divided into four groups according to the position. The number of cylinders in Groups A and C is 4, and the number of cylinders in Groups B and D is 7. The cylinder section is shown in Figure 16b.

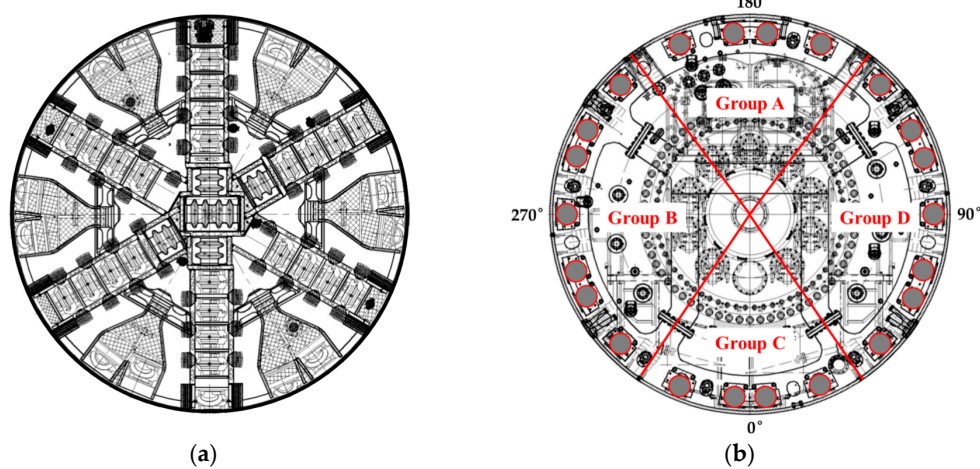

(**a**)　　　　　　　　　　　　　　　　(**b**)

**Figure 16.** Cutterhead and advance system of shield: (**a**) shield cutterhead; and (**b**) shield cylinder partition.

**Table 2.** Summary of the main specifications for the EPB.

| Shield Type | EPB |
| --- | --- |
| External diameter (m) | 6.68 |
| The length of shield (m) | 9.0 |
| Shield weight (t) | Approximately 500 |
| Shield eccentricity (m) | Approximately 0.5 |
| Cutter opening ratio (%) | 35 |
| Maximum total thrust (kN) | 40,860 |
| Number of propulsion cylinders | 22 |
| Maximum advance speed (mm/min) | 80 |

## 5. Results and Discussion

### 5.1. Engineering Application I: Inversion Calculation of Soil-Shield Interaction Force

The soil layer above the section of Ring 504 is composed of backfill and loess, which is a non-uniform clay stratum. According to the method of this paper and in [27], initial force acting on the shield periphery can be calculated in the non-uniform clay stratum. As shown in Figure 15, the section of Ring 504 is also a non-uniform layer. The loess layer and silty clay each account for about half of the entire excavation surface and can be approximated by 1:1.

Through the guiding system and data monitoring system of the EPB, when the shield had been advancing at Ring 504, partition cylinder thrust, pitch angle and yawing angle were recorded every 1 min. The time to excavate Ring 504 was about 50 min, but for various reasons, it was interrupted during the tunneling process. Continuous monitoring data were more valuable and informative. The longest continuous boring time of the ring was about 15 min. To avoid the influence of shield start and stop, this study selected continuous monitoring data of 10 min in the middle of this time. The data for a period were selected for analysis and comparison. The measured data of the pitch angle and the yawing angle are shown in Figure 17. The yawing angle varied greatly, while the pitch angle remained basically stable.

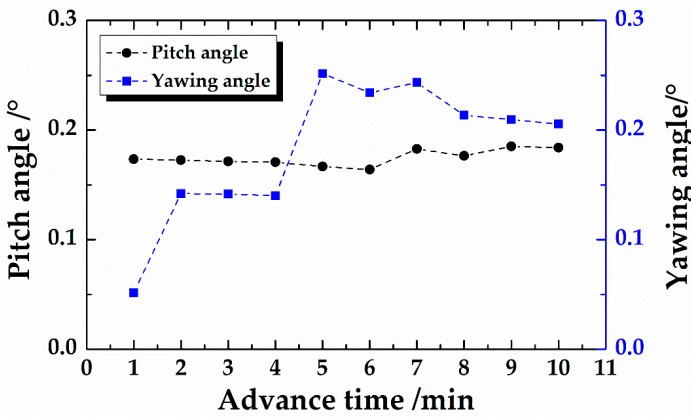

**Figure 17.** Pitch angle and yawing angle measured data.

The pitch angle and the yawing angle had been known through the guiding system. According to the flow chart of the shield attitude calculation (Figure 11), the force of shield–soil interaction during the tunneling process can be inverted. The calculation results are shown in Figure 18. Figure 18 shows the distribution of the total normal earth pressure around the shield periphery. The height of graphs represents the length of shield, the width of graphs represents the shield periphery from 0° to 360°. Along the line at 180°, it can be found that the normal pressure is gradually decreasing from the bottom to the top of graphs, indicating that the pitch angle of the shield is positive. The gradient of the normal pressure reduction reflects the magnitude of the pitch angle. The stress nephogram on the left or right of the line at 180° shows the direction of the yawing angle of the shield.

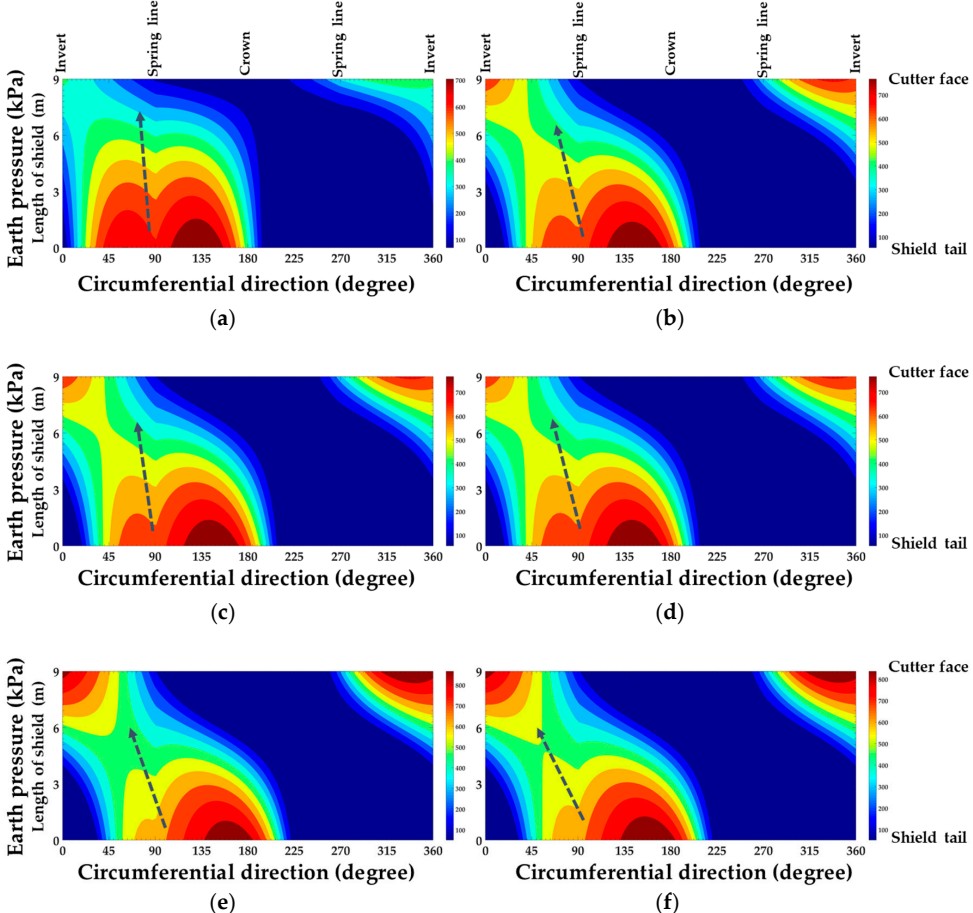

**Figure 18.** *Cont.*

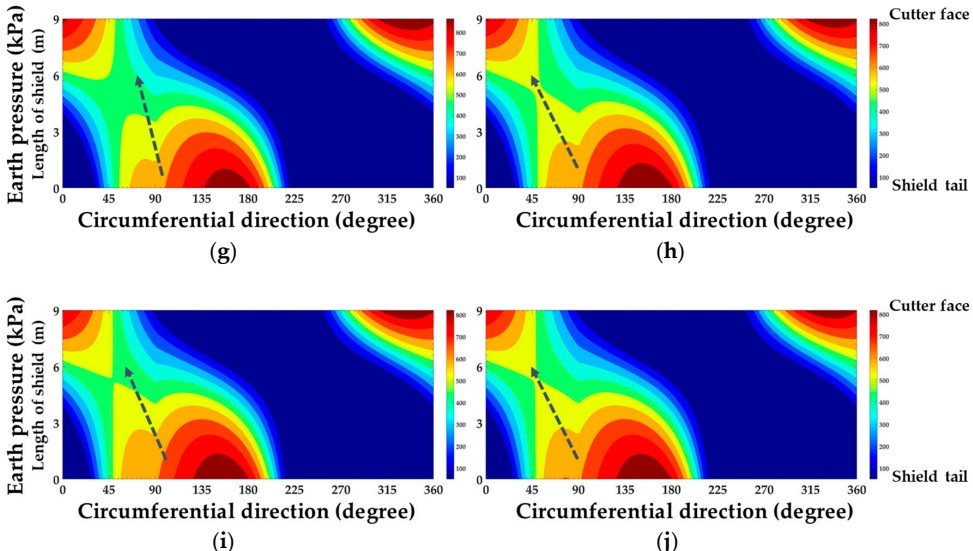

**Figure 18.** Shield stress nephogram at each moment: (**a**) 1 min; (**b**) 2 min; (**c**) 3 min; (**d**) 4 min; (**e**) 5 min; (**f**) 6 min; (**g**) 7 min; (**h**) 8 min; (**i**) 9 min; and (**j**) 10 min.

The change of the pitch angle of the shield is small, and the fluctuation of the stress nephogram is mainly due to the change of the yawing angle. The stress nephogram can be used to preliminarily determine the degree and the direction of deflection on the horizontal plane of the shield. The greater is the yawing angle, the more severe is the distortion of the stress nephogram. For example, there was a sudden increase of the yawing angle between time = 4 min and time = 5 min, as can be seen from the stress nephogram.

*5.2. Engineering Application II: Shield Pitch Angle Prediction*

The second engineering application is the prediction of the pitch angle. Combined with the Jinan Metro Line R2, when the advance force applied by the four cylinder partitions is known, the attitude change of the shield can be predicted in advance. The soil mass $W_{soil}$ with the same volume as the shield is about 600 t, which is larger than the self-weight of the shield machine. Therefore, under the premise of not applying any external force, there will be a phenomenon of "upward floating", that is, $\Delta s_v < 0$. The first step is to calculate $\Delta s_v$ ($\Delta s_v = -13$ mm), according to the method in Figure 11.

The $M_A^\beta$ actively applied by the EPB can be calculated by the thrust difference between Group A and Group C of the advance system. The $M_G$ is 2.5 MN·m, which can be calculated. The change curve of the $M_A^\beta + M_G$ can be obtained during construction, as shown in Figure 19. When $M_A^\beta + M_G$ and $\Delta s_v$ are known, the theoretical values of the shield pitch angle during the study period can be obtained. The theoretical values are calculated by the $M_A^\beta + M_G$. Therefore, the trend of theoretical values should be similar to the trend of the $M_A^\beta + M_G$. However, the measured values did not have this trend, indicating that the $M_A^\beta + M_G$ did not fully act on shield attitude adjustment.

Figure 20 is a comparative analysis of the measured and predicted values of the shield pitch angle during the study period. It can be seen in the figure that the theoretical value is generally larger than the measured value, indicating that the moment actively applied by the shield through the jack cylinder cannot fully act on the shield itself. When the advance cylinder acts on the shield, the friction at the joint and the characteristics of the shield itself may cause the deflection moment to decrease, resulting in a smaller pitch angle actually being formed.

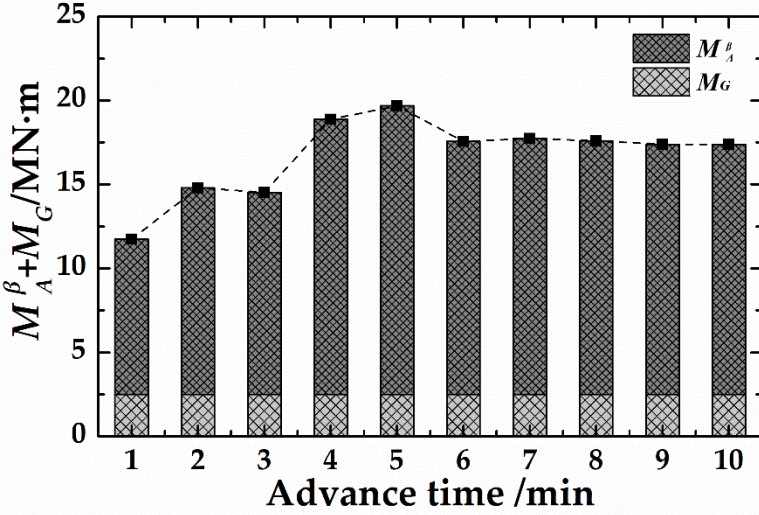

**Figure 19.** The change graph of $M_A^\beta + M_G$.

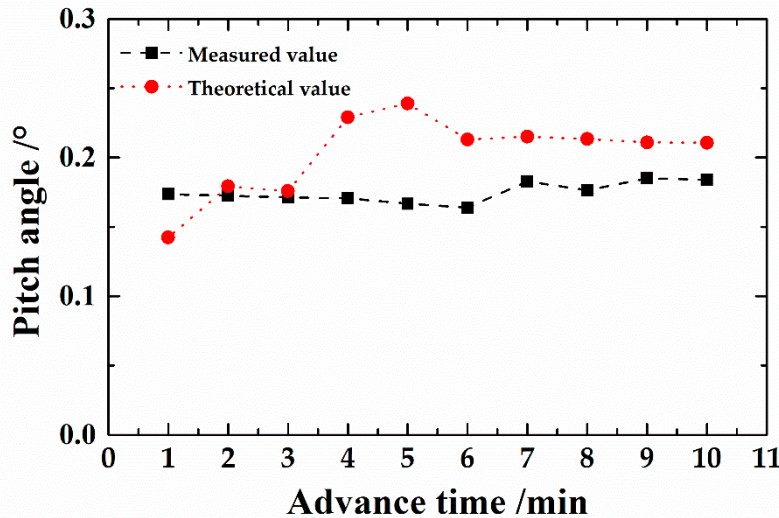

**Figure 20.** Contrast between the measured and the theoretical value of the shield pitch angle.

*5.3. Engineering Application III: Shield Yawing Angle Prediction*

The third engineering application is the prediction of the yawing angle. The $\Delta s_v$ is calculated in the first stage. The theoretical value of the pitch angle at each moment is shown in Figure 20. The known $\Delta s_v$ and $\beta$ are brought into the calculation process of the third stage to obtain the RCYH (Figure 21). The $M_A^\alpha$ actively applied by the EPB can be calculated by the thrust difference between Group B and Group D of the advance system. The change curve of the $M_A^\alpha$ is shown in Figure 22. Then, the theoretical value of the yawing angle can be obtained. The comparison between the theoretical value and the measured value is shown in Figure 23. The trends of theoretical and measured values are both similar to the trend of the $M_A^\alpha$. The $M_A^\alpha$ is obtained by measurement. It indicates that the $M_A^\alpha$ applied by the shield can act more fully on the shield than $M_A^\beta$.

Comparing the measured values with the theoretical values in Figure 23, it can be found that the trend of the theoretical value is basically similar to the measured value, but the theoretical value is generally larger than the measured value. In the calculation of the theoretical value, the deflection moment value is the initial data, and the moment used for the shield correction is reduced due to mechanical loss, etc., so that the final yawing angle formed by the shield is small.

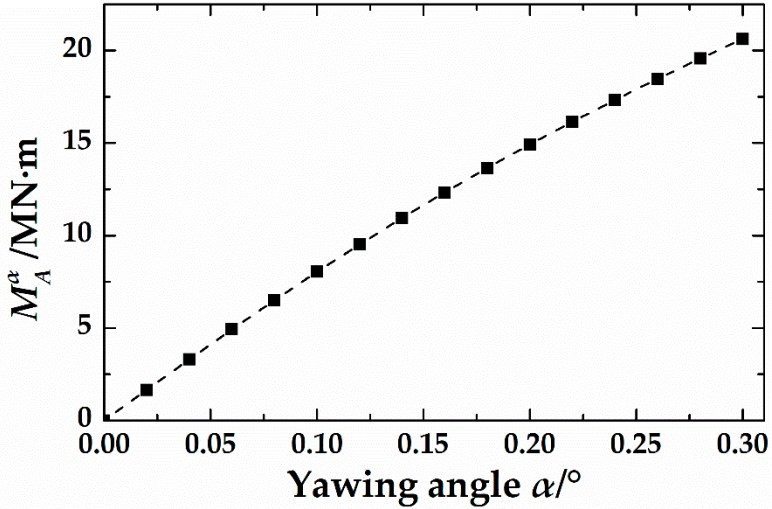

**Figure 21.** The relationship between the yawing angle and $M_A^\alpha$.

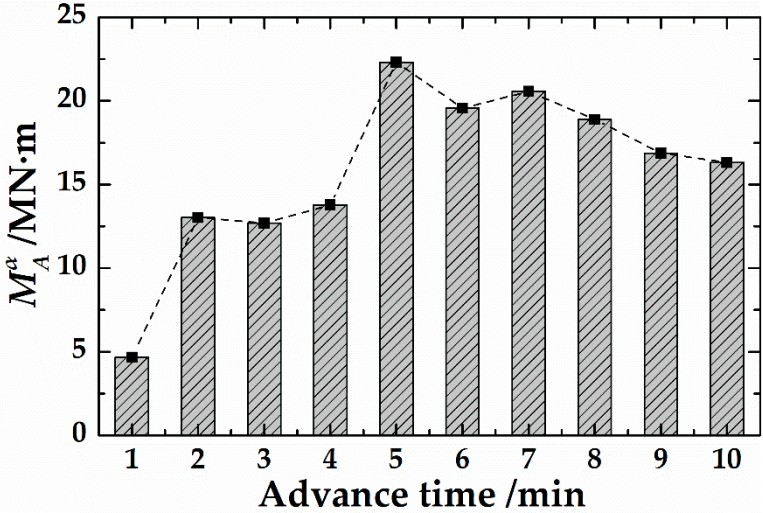

**Figure 22.** The change graph of $M_A^\alpha$.

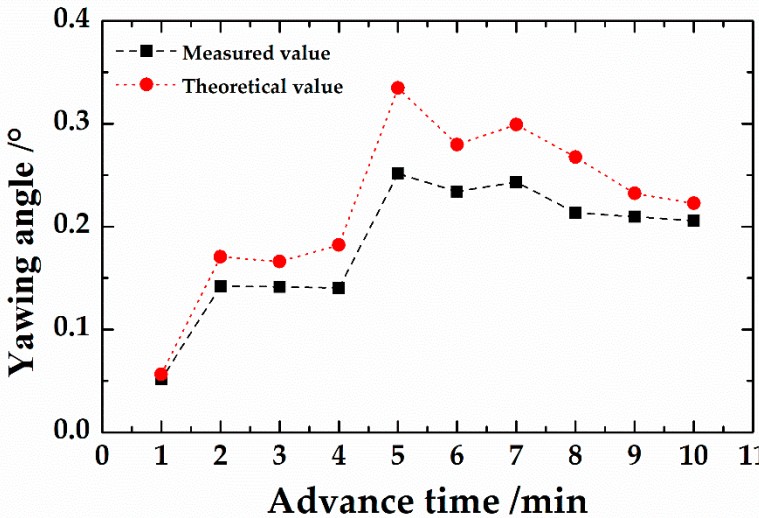

**Figure 23.** Contrast between the measured and the theoretical value of the shield yawing angle.

## 6. Conclusions

This study revealed the mechanism of the interaction between shield and soil during the change of shield attitude. Through theoretical analysis, programming calculation and field monitoring, the calculation method of the pitch angle and yawing angle of the shield was studied and verified. The main achievements of this study are outlined as follows:

(1) Based on the ground reaction force curve, the interaction between the shield and the soil was simulated by the equivalent spring, and the theoretical calculation method of $f_5$ was obtained in the change of the shield attitude.

(2) The improved calculation method of loose earth pressure solved the initial boundary problem of the shield attitude calculation. Combined with the theoretical calculation method of $f_5$, the calculation process of the shield attitude was formed.

(3) Based on the monitoring data of Jinan Metro Line R2, the interaction between the shield and the soil during the construction of the shield was inverted. Through the stress nephogram, the stress concentration area of the shield can be judged to guide the next step of attitude adjustment.

(4) Based on the measured data of the project, the theoretical calculation method of the pitch angle and yawing angle of the shield was verified. The study found that the theoretical value was close to the measured value, but, since the moment generated by the advance cylinders cannot fully act on the shield itself, the theoretical values of the shield attitude angles were generally greater than the measured values.

(5) The model of shield–soil interaction has some benefits for upcoming projects. Firstly, this model can guide the shield operator to perform shield attitude correction. Secondly, it is possible to initially obtain a position where the attitude control of any shield tunnel construction is difficult for upcoming projects.

**Author Contributions:** X.S., D.-J.Y. and D.-L.J. contributed equally to this work.

**Funding:** The research work described herein was funded by the National Basic Research Program of China (973 Program: 2015CB057800) and Fundamental Research Funds for the Central Universities of China (NO. 2017YJS151).

**Acknowledgments:** The research work described herein was funded by the National Basic Research Program of China. This financial support is gratefully acknowledged.

**Conflicts of Interest:** The authors declare no conflict of interest.

## Notations

| Symbol | Description | Unit | Symbol | Description | Unit |
|---|---|---|---|---|---|
| $f_1$ | Shield self-weight | kPa | $f_2$ | Shield tail load | kPa |
| $f_3$ | Advance load provided by the jacks | kPa | $f_4$ | Load acting on the cutterhead | kPa |
| $f_5$ | Force acting on the shield periphery | kPa | $(x_0, y_0, z_0)$ | cutterhead center coordinates | m |
| $\alpha$ | Yawing angle | ° | $\beta$ | Pitch angle | ° |
| $\Omega$ | Rolling angle | ° | $C$ | Coordinate system | [-] |
| $F$ | Total force | kN | $M$ | Total moment | MN·m |
| $\sigma_v$ | Loose earth pressure | kPa | $K_0$ | Later pressure coefficient | [-] |
| $H$ | Overlying soil thickness | m | $P_0$ | Upper load | kPa |
| $c$ | Cohesion | kPa | $\varphi$ | Internal friction angle | ° |
| $\gamma$ | Weight density | kN/m$^3$ | $B_1$ | Loose band width | m |
| $R$ | Radius of the tunnel | m | $A_1$ | Characteristic parameters | [-] |
| $p_1$ | Initial soil pressure upper value | kPa | $p_2$ | Initial earth pressure lower value | kPa |
| $D$ | Shield diameter | m | $L$ | Shield length | m |
| $l_s$ | Shield eccentricity | m | $\Delta s$ | Shield displacement | m |
| $U$ | Shield shell unit displacement | m | $\eta$ | Angle with the $p$-axis in the $C^M$ | ° |
| $K$ | coefficient of earth pressure | [-] | $a$ | the gradient of functions $K(U)$ | [-] |
| $\sigma$ | Stress | kPa | $A$ | Area | m$^2$ |
| **Superscripts** | | | | | |
| $E$ | Global coordinate system | [-] | $M$ | Machine coordinate system | [-] |
| $MR$ | Rotated coordinate system | [-] | $b$ | Different coordinate systems | [-] |

**Subscripts**

| | | | | | |
|---|---|---|---|---|---|
| *h, v* | Horizontal and vertical directions | [-] | *e* | Different directions of force | [-] |
| *A* | Shield apply | [-] | *max* | Maximum | [-] |
| *min* | Minimum | [-] | *o* | Origin or initial | [-] |
| *i, j* | *i*th and *j*th calculation points | [-] | *s* | Final | [-] |

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
