# Peer review of "Influence of Shield Attitude Change on Shield–Soil Interaction"

_applsci, doi:10.3390/app9091812_

Round 1
Reviewer 1 Report
Chapter 2) descirption of methodology remains insufficient. Is is a linear or non-linear approach?
Figure 18 is not selfexplaining. Please add a detailed discussion.
Figure 20: what is the reason for the "jump" within the theoretical values between 4 - 5 advance time/min?
Figure 23: what is the reason for the "jump" within the theoretical + measured values between 4 - 5 advance time/min?
Conclusion: Please explain the benefit of the model for upcoming projects.
Author Response
Point 1: Chapter 2) descirption of methodology remains insufficient. Is a linear or non-linear approach?
Response 1: The interaction between the shield and the soil is approximated by a nonlinear spring. The coefficient of the nonlinear spring can be obtained from the ground reaction curve
I have added the relevant description in the second chapter.
Point 2: Figure 18 is not selfexplaining. Please add a detailed discussion
Response 2: Figure 18 shows the distribution of the total normal earth pressure around the shield periphery. The height of graphs represents the length of shield, the width of graphs represents the shield periphery from 0° to 360°. Along the line at 180°, it can be found that the normal pressure is gradually decreasing from the bottom to the top of graphs, indicating that the pitch angle of the shield is positive. The gradient of the normal pressure reduction reflects the magnitude of the pitch angle. The stress nephogram on the left or right of the line at 180° shows the direction of the yawing angle of the shield. The greater the yawing angle, the more severe the distortion of the stress nephogram.
I have added relevant content to the text in Line 295~301.
Point 3: Figure 20: what is the reason for the "jump" within the theoretical values between 4 - 5 advance time/min?
Response 3: The adjustment of the shield attitude is carried out through four groups of cylinders. The change in the pitch angle of the shield is due to the MG and the deflection moment generated by the pressure difference between the two groups (A and C). As can be seen from Figure 19, there was a sudden increase of the deflection moment MβA between 4 - 5 advance time/min. The theoretical values of pitch angle are calculated by MβA+MG. Therefore the trend of theoretical values should be similar to the trend of the MβA+MG. But the measured values did not have this trend, indicating that the MβA+MG did not fully act on shield attitude adjustment. So the theoretical values have a “jump” between 4 - 5 advance time/min.
I have added relevant content to the text in Line 323~325.
Point 4: Figure 23: what is the reason for the "jump" within the theoretical + measured values between 4 - 5 advance time/min?
Response 4: Figure 21 shows the relationship the yawing angle and MαA. The theoretical value of the yawing angle under the is obtained from Figure 21. Therefore, the trend of theoretical values should be similar to the trend of the MαA. And the MαA is calculated by measurement.
The adjustment of the shield attitude is carried out through four groups of cylinders. The change in the yaw angle of the shield is due to the deflection moment generated by the pressure difference between the two groups (B and D). As can be seen from Figure 22, there was a sudden increase of the deflection moment MαA between 4 - 5 advance time/min. The trend of theoretical and measured values are both similar to the trend of the MαA. And the MαA is calculated by measurement. This indicates that the MαA applied by the shield can act more fully on the shield than MβA.
So the theoretical + measured values have a “jump” between 4 - 5 advance time/min.
I have added relevant content to the text in Line 343~345.
Point 5: Conclusion: Please explain the benefit of the model for upcoming projects.
Responds to comment:
Response 5: (1) This model can guide the shield operator to perform shield attitude correction;
(2) Using this model, it is possible to initially obtain a position where the attitude control of any shield tunnel construction is difficult for upcoming projects;
I have added relevant content to the text in Line 377~380.
Note: The number of line is indicated under the "Track Changes" function in Microsoft Word
I would like to express my sincere gratitude to the editors and the reviewer for their hard work!
Sincerely!

Reviewer 2 Report
The paper presents a well based analysis of the mechanism of the shield-soil interaction during the change of shield attitude. It represents a step forward in the field, developing a method for calculation and verified with field data.
Some corrections must to be made in order to be accepted for publication:
Lines 264-276. The main question is why the analysis period is limited to 10 minutes? This must be explained in the text. If more data are available, why are not they used?
Lines 120-140. Provide a reference for a further explanation of the phases.
Lines 72-73. Please, include a figure for indicating the yawing angle, the pitch angle and the rolling angle.
Line 104. Although η can be identified with an angle, please, define it when introducing it for the first time.
Line 207. Correct the sentence: “Suppose a shield tunnel is excavated in A uniform sand layer”
Line 231. Correct the sentence: The buried depth of the tunnel VARIED from 7.9 to 15.2 m.
Line 232. (The line) underneath pass through important buildings such as high-speed railway, expressway and River. Are they buildings? Correct it.
Line 236. What is Ring 504? What does it mean/represent?
Figure 13. The figure at the left is supposed to be China. Please, indicate it, and include neighbor countries. Additionally, a greater image of the town of Jinan and existing metro lines is recommended, better in color. The legend of the figure is not correct; it does not correspond to the figure.
Figure 14. The legend of the figure does not indicate the real one.
Line 250. It must refer to Table 2.
Line 261. The expression “…according to the method of Literature [23]” is not correct. Rewrite it.
On the other hand, with regard to cites in the text, there is repetitive mistake. There are various ways of citing an article. One of them is by means of an active sentence, indicating that Someone has proposed a model [5]. This is a usual form. But, normally, if more than one reference is included with the name of the authors, all the references included must be authored by the same researchers.
For example, in line 39 it is indicated that “Koyama et al [6,7] applied the automatic trajectory…”. However, when looking to the references, the sixth one is authored by Koyama (as the unique author) and the seventh one by other authors. This Is completely incorrect. On one hand, there is a paper by Koyama and, on the other hand, another one by Xie, Duan, Yang and Liu. This must be corrected. Moreover, do different authors do the same? It is not clear after reading the sentence.
A similar example can be found in line 48: “Yue [12,13] proposed a shield attitude…” However, although Yue appears in references 12 and 13, the first one would be indicated as Yue et al. and the second one as Sun et al. A correction is needed.
Line 50: Ates et al. [14,15]. Ates et al. only refers to reference 14.
Line 42. “Shimizu et al. [8,9]. In this case, both papers should be indicated in the text as Shimizu et al., but the other authors are not the same ones. The use of a unique subject when referring to more than one paper is not recommended. The same occurs in line 61, “Sugimoto [16,17]. Both papers are authored by Sugimoto, but the coauthors are not the same and they are omitted.
Author Response
Point 1: Lines 264-276. The main question is why the analysis period is limited to 10 minutes? This must be explained in the text. If more data are available, why are not they used?
Response 1: Thanks for your suggestion. Explain as follows:
The time to excavate the Ring.504 was about 50 minutes, but for various reasons, it was interrupted during the tunneling process. Continuous monitoring data were more valuable and informative. The longest continuous boring time of the ring was about 15 minutes. In order to avoid the influence of shield start and stop, this paper selected continuous monitoring data of 10 minutes in the middle of this time.
Related content is seen in Line 286~290
Point 2: Lines 120-140. Provide a reference for a further explanation of the phases.
Response 2: Thanks for pointing the error. The relevant explanations and references for the assumptions are missing in the paper.
When the shield attitude is adjusted in the actual project, the yawing angle and pitch angle should be changed simultaneously. However, when the time interval of shield attitude adjustment tends to infinitesimal, it can be considered that the pitch angle and yawing angle are in succession. And the order in which the pitch and yawing angles occur does not affect the final calculation result from the aspect of the mechanics analysis. So I made the assumptions in the paper.
Related content is seen in Line 133~137.
Point 3: Lines 72-73. Please, include a figure for indicating the yawing angle, the pitch angle and the rolling angle.
Response 3: In response to the comments made by the reviewer, it is necessary to add a picture to clearly show the yawing angle, pitch angle and rolling angle. It is shown in Figure 2(a). Related content is marked in Line 75.
Point 4: Line 104. Although η can be identified with an angle, please, define it when introducing it for the first time.
Response 4: η is the angle with the p-axis in the CM coordinate system. Related content is marked in Line 110.
Point 5: Line 207. Correct the sentence: “Suppose a shield tunnel is excavated in A uniform sand layer”
Response 5: Thanks for pointing the error. It has been corrected in Line 222
Point 6: Line 231. Correct the sentence: The buried depth of the tunnel VARIED from 7.9 to 15.2 m.
Response 6: Thanks for pointing the error. It has been corrected in Line 246
Point 7: Line 232. (The line) underneath pass through important buildings such as high-speed railway, expressway and River. Are they buildings? Correct it.
Response 7: Thanks for pointing the error. “buildings” is replaced by “regions” in Line 247.
Point 8: Line 236. What is Ring 504? What does it mean/represent?
Response 8: Ring 504 means the 504th ring segment. It has been added in Line 251~252
Point 9: Figure 13. The figure at the left is supposed to be China. Please, indicate it, and include neighbor countries. Additionally, a greater image of the town of Jinan and existing metro lines is recommended, better in color. The legend of the figure is not correct; it does not correspond to the figure.
Response 9: Thanks for pointing the error. China and neighbor countries have been indicated in the new figure. Metro Line 2 has been added into the image of the town of Jinan which is in color. The legend of the figure has been corrected.
Point 10: Figure 14. The legend of the figure does not indicate the real one.
Response 10: Thanks for pointing the error. I removed the irrelevant information from the graph. Different soil layers are represented by different legends and marked below the figure.
Point 11: Line 250. It must refer to Table 2.
Response 11: Thanks for pointing the error. It has been corrected in Line 268
Point 12: Line 261. The expression “…according to the method of Literature [23]” is not correct. Rewrite it.
Response 12: The sentence is modified as follows: According to the method of this paper and Literature [23], initial force acting on the shield periphery can be calculated in the non-uniform clay stratum. Related content is marked in Line 278~281.
Point 13: On the other hand, with regard to cites in the text, there is repetitive mistake. There are various ways of citing an article. One of them is by means of an active sentence, indicating that Someone has proposed a model [5]. This is a usual form. But, normally, if more than one reference is included with the name of the authors, all the references included must be authored by the same researchers.
Response 13: Thanks for your point the error. I have carefully checked the Chapter 1 based on your opinion.
(1) For example, in line 39 it is indicated that “Koyama et al [6,7] applied the automatic trajectory…”. However, when looking to the references, the sixth one is authored by Koyama (as the unique author) and the seventh one by other authors. This Is completely incorrect. On one hand, there is a paper by Koyama and, on the other hand, another one by Xie, Duan, Yang and Liu. This must be corrected. Moreover, do different authors do the same? It is not clear after reading the sentence.
Response 13(1): Thanks for pointing the mistake. Xie et al. [6] applied the automatic trajectory tracking control system to the control of the shield's route. Koyama’s paper (Present status and technology of shield tunneling method in Japan) is a literature review. The sentence “these systems only relied on empirical relationships, without precise theoretical background, and it was impossible to deal with the precise control of the shield machine in the case of complex geological structures and sharp curves” is mentioned in this paper.
(2) A similar example can be found in line 48: “Yue [12,13] proposed a shield attitude…” However, although Yue appears in references 12 and 13, the first one would be indicated as Yue et al. and the second one as Sun et al. A correction is needed.
Response 13(2): Thanks for pointing the mistake. The error has been corrected in the text.
(3) Line 50: Ates et al. [14,15]. Ates et al. only refers to reference 14.
Response 13(3): Ates and Zhang respectively proposed different methods for predicting the thrust and torque requirements of different shield machines. The error has been corrected in the text.
(4) Line 42. “Shimizu et al. [8,9]. In this case, both papers should be indicated in the text as Shimizu et al., but the other authors are not the same ones. The use of a unique subject when referring to more than one paper is not recommended. The same occurs in line 61, “Sugimoto [16,17]. Both papers are authored by Sugimoto, but the coauthors are not the same and they are omitted.
Response 13(4): Shimizu and Isaka [8] analyzed the tunneling and motion laws of tunnel construction without the influence of nonlinear factors, and Shimizu and Obayashi et al. [9] established a mathematical analysis model of shield tunneling movement.
The literature 17 has been removed in Line 63. And the Literature 17 is cited in line 69.
Note: The number of line is indicated under the "Track Changes" function in Microsoft Word
I would like to express my sincere gratitude to the editors and the reviewer for their hard work!
Sincerely!

Round 2
Reviewer 2 Report
The revised version of the paper explains better some of the indicated points. Sections 2, 3 and 4 expose more clearly the presented procedure. However, the Introduction could be too short and does not give a global overview of the literature about Tunneling Boring Machines (TBM). With regard to this point, it would worth to mention some references of the literature about proposed prediction models of the TBM performance, depending on different factors as Unconfined Compressive Strength, RQD, diameter of the machine or properties of the cutting edges. Even a mention about prediction models for Micro Tunneling Boring Machines (MTBM) could be useful. This could be the most interesting point to revise.
On the other hand, although some corrections had been conducted, some confusion remains. In line 39 it is indicated that “Xie et al. [6] applied the automatic trajectory tracking...and it was impossible to deal with the precise control of the shield machine in the case of complex geological structures and sharp curves [7]. If Xie et al. [6] applied a new procedure, it is strange that the conclusions of it are included in another reference, number 7, when, moreover, reference 7 is from 2003 and number 6 is from 2012. It seems extremely strange. These references are requested to be revised.
With regard to references 8 and 9, although when including in the text they abbreviated similarly (Shimizu et al.), it is not necessary to include the second author, as they are perfectly referred with numbers (8 and 9, respectively).
Finally, a further description of the proposed different methods by Ates et al. [14] and Zhang [15] would improve the quality of the paper.
Author Response
Point 1: The revised version of the paper explains better some of the indicated points. Sections 2, 3 and 4 expose more clearly the presented procedure. However, the Introduction could be too short and does not give a global overview of the literature about Tunneling Boring Machines (TBM). With regard to this point, it would worth to mention some references of the literature about proposed prediction models of the TBM performance, depending on different factors as Unconfined Compressive Strength, RQD, diameter of the machine or properties of the cutting edges. Even a mention about prediction models for Micro Tunneling Boring Machines (MTBM) could be useful. This could be the most interesting point to revise.
Response 1: Thanks for the guidance of the paper. I have added related literature into introductions
With the necessity to accurately predict performance of machines in different ground conditions, many researchers have worked to develop new prediction models or adjustment factors for the common existing models [16]. Acaroglu et al [17] established a model to predict specific energy requirement of constant cross-section disc cutters in the rock cutting process by using fuzzy logic method. Barton et al [18] proposed a model named QTBM which uses many input parameters (such as RQD, joint condition, Stress condition, intact rock strength, quartz content and TBM thrust). The modified CSM model was added rock mass properties as input parameters into the model [19]. In hard rock formations, such models have had many research results no longer expand too much. However, in the soft soil layer, the interaction model for shield tunneling is still in its infancy.
Point 2: On the other hand, although some corrections had been conducted, some confusion remains. In line 39 it is indicated that “Xie et al. [6] applied the automatic trajectory tracking...and it was impossible to deal with the precise control of the shield machine in the case of complex geological structures and sharp curves [7]. If Xie et al. [6] applied a new procedure, it is strange that the conclusions of it are included in another reference, number 7, when, moreover, reference 7 is from 2003 and number 6 is from 2012. It seems extremely strange. These references are requested to be revised.
Response 2: Thanks for pointing the error. Make the following modifications:
Koyama et al. [6] mentioned that it was impossible to deal with the precise control of the shield machine in the case of complex geological structures and sharp curves. Xie et al. [7] applied the automatic trajectory tracking control system to the control of the shield's route, but these systems also only relied on empirical relationships, without precise theoretical background.
Point 3: With regard to references 8 and 9, although when including in the text they abbreviated similarly (Shimizu et al.), it is not necessary to include the second author, as they are perfectly referred with numbers (8 and 9, respectively).
Response 3: Thanks for pointing the error. It has been modified in the paper.
Point 4: Finally, a further description of the proposed different methods by Ates et al. [14] and Zhang [15] would improve the quality of the paper.
Response 4: Thanks for the guidance of the paper. Make the following modifications:
Ates et al [14] thought that it’s crucially important to select a proper TBM and define its basic specifications such as installed cutter-head torque and TBM thrust capacities. Zhang et al [15] established a predicting model of the thrust and torque for the total load that fully reflects the influences of geological, operating, and structural parameters.
